# A Diagnostic Strategy for Gauging Individual Humoral Ex Vivo Immune Responsiveness Following COVID-19 Vaccination

**DOI:** 10.3390/vaccines10071044

**Published:** 2022-06-29

**Authors:** Anna Sabrina Kuechler, Sandra Weinhold, Fritz Boege, Ortwin Adams, Lisa Müller, Florian Babor, Sabrina B. Bennstein, T.-X. Uyen Pham, Maryam Hejazi, Sarah B. Reusing, Derik Hermsen, Markus Uhrberg, Karin Schulze-Bosse

**Affiliations:** 1Central Institute for Clinical Chemistry and Laboratory Diagnostics, Medical Faculty, University Hospital Düsseldorf, Heinrich-Heine-University, 40225 Düsseldorf, Germany; boege@med.uni-duesseldorf.de (F.B.); hermsen@med.uni-duesseldorf.de (D.H.); karin.schulze-bosse@med.uni-duesseldorf.de (K.S.-B.); 2Institute for Transplantation Diagnostics and Cell Therapeutics, University Hospital Düsseldorf, Heinrich-Heine-University, 40225 Düsseldorf, Germany; sandra.weinhold@med.uni-duesseldorf.de (S.W.); sabrinabianca.bennstein@med.uni-duesseldorf.de (S.B.B.); txuanuyen@gmail.com (T.-X.U.P.); maryam79@gmx.de (M.H.); sarah_reusing@web.de (S.B.R.); markus.uhrberg@med.uni-duesseldorf.de (M.U.); 3Institute of Virology, Medical Faculty, University Hospital Düsseldorf, Heinrich-Heine-University, 40225 Düsseldorf, Germany; ortwin.adams@med.uni-duesseldorf.de (O.A.); lisa.mueller2@med.uni-duesseldorf.de (L.M.); 4Institute of Hematology, Oncology and Clinical Immunology, Center for Child and Adolescent Health, University Hospital Düsseldorf, Heinrich-Heine-University, 40225 Düsseldorf, Germany; florian.babor@med.uni-duesseldorf.de

**Keywords:** COVID-19 serology, SARS-CoV-2 neutralization, SARS-CoV-2 vaccination, SARS-CoV-2 immunity, companion diagnostic

## Abstract

Purpose: We describe a diagnostic procedure suitable for scheduling (re-)vaccination against severe acute respiratory syndrome coronavirus type 2 (SARS-CoV-2) according to individual state of humoral immunization. Methods: To clarify the relation between quantitative antibody measurements and humoral ex vivo immune responsiveness, we monitored 124 individuals before, during and six months after vaccination with Spikevax (Moderna, Cambridge, MA, USA). Antibodies against SARS-CoV-2 spike (S1) protein receptor-binding domain (S1-AB) and against nucleocapsid antigens were measured by chemiluminescent immunoassay (Roche). Virus-neutralizing activities were determined by surrogate assays (NeutraLISA, Euroimmune; cPass, GenScript). Neutralization of SARS-CoV-2 in cell culture (full virus NT) served as an ex vivo correlate for humoral immune responsiveness. Results: Vaccination responses varied considerably. Six months after the second vaccination, participants still positive for the full virus NT were safely determined by S1-AB levels ≥1000 U/mL. The full virus NT-positive fraction of participants with S1-AB levels <1000 U/mL was identified by virus-neutralizing activities >70% as determined by surrogate assays (NeutraLISA or cPas). Participants that were full virus NT-negative and presumably insufficiently protected could thus be identified by a sensitivity of >83% and a specificity of >95%. Conclusion: The described diagnostic strategy possibly supports individualized (re-)vaccination schedules based on simple and rapid measurement of serum-based SARS-CoV-2 antibody levels. Our data apply only to WUHAN-type SARS-CoV-2 virus and the current version of the mRNA vaccine from Moderna (Cambridge, MA, USA). Adaptation to other vaccines and more recent SARS-CoV-2 strains will require modification of cut-offs and re-evaluation of sensitivity/specificity.

## 1. Introduction

Since the emergence of severe acute respiratory syndrome coronavirus type 2 (SARS-CoV-2) in December 2019, more than 300,000,000 cases have been reported globally and nearly six million deaths have been confirmed [1,2]. The virus is transmitted human-to-human and can affect almost all human organs causing COVID-19, a potentially chronic disease comprising, inter alia, dry cough, fever, dyspnoea, anosmia, ageusia and pneumonia [2,3]. Apart from anti-inflammatory and virostatic approaches for treating manifest COVID-19, vaccination is considered the most crucial measure for stopping the SARS-CoV-2-pandemic [4].

SARS-CoV-2 constitutes a new scientific problem. Over the past two years, many questions have been addressed regarding disease management, and vaccination very quickly came into focus. The type of vaccine, which part of the virus should be targeted by vaccination and which immunization schemes should be employed had to be determined [5]. Several types of vaccines were developed rapidly, but initially limited supply of doses mandated decisions on whom to vaccinate first [6]. By January 2022, nearly ten billion doses had been administered globally [1], and the scientific focus shifted towards surveillance of individual vaccination responses and optimization of renewed immunization.

In the latter context, serological tests for monitoring humoral immune responses to infection and/or vaccination are of central importance. In December 2020, the World Health Organization established an international standard and reference panel allowing calibrated and standardized determinations of circulating levels of SARS-CoV-2 antibodies [7,8]. However, scientific guidance is lacking regarding which levels of circulating antibodies are sufficient for protecting vaccinated patients from severe causes of an infection and which serological tests are suited for surveillance of SARS-CoV-2 immune responsiveness following infections or vaccinations [7].

Here, we investigate how the state of humoral immune responsiveness against SARS-CoV-2 following vaccination (as specified by circulating serum levels of specific antibodies and functional ex vivo effects thereof) can be assessed under conditions of routine health care. We monitored the vaccination response of an adult cohort during two cycles of vaccination with the mRNA-based COVID-19 vaccine Spikevax (Moderna Biotech, Cambridge, MA, USA), using immunoassays for antibodies against the SARS-CoV-2 spike (S1) protein receptor-binding domain (S1-AB) and the SARS-CoV-2 nucleocapsid antigen (N-AB) and two surrogate assays for the virus-neutralizing activity of SARS-CoV-2 antibodies, for which measurements of suppression of cytopathic effects of the SARS-CoV-2 virus in cell culture (full virus neutralization test, full virus NT) served as a reference. 

We thus evaluated a set of routine serological tests for circulating SARS-CoV-2 antibodies as COVID-19 companion diagnostics and aimed to establish criteria for judging their results quoad functional ex vivo responsiveness against the SARS-CoV-2 virus. Based on our results, we propose a staged diagnostic strategy that may allow laboratories to monitor the functional state of humoral immune responsiveness to SARS-CoV-2, without having access to a BSL-3 facility required for the full virus neutralization test, which is considered the gold standard.

## 2. Materials and Methods

### 2.1. Study Participants

A total of 124 study participants (83 female, 41 male, mean age 46 years, median age of 50 years) were recruited at the University Hospital of the Heinrich Heine University, Düsseldorf. All participants were employees of these institutions and underwent a program of two vaccinations with the COVID-19 vaccine Spikevax (Moderna Biotech, Cambridge, MA, USA) spaced exactly four weeks apart. Vaccinations were performed according to the instructions of the manufacturer and the recommendations of the German vaccination commission (STIKO). None of the participants tested positive for SARS-CoV-2 or exhibited symptoms of COVID-19, or exhibited debilitating symptoms of co-morbidities.

### 2.2. Sampling

Blood samples (18 mL) were collected by antecubital vein puncture 48 h before and four weeks following initial vaccination, and two weeks and six months after the booster dose (i.e., the second dose). Immune responses to mRNA-based vaccines are known to be reliably detectable 21 days after initial and seven days after a booster dose [9]. Following centrifugation (20 min, 1650× *g*), serum was separated and stored at −20 °C until testing. Aliquots (1 mL) for reflex testing were stored at −20 °C for up to 6 months.

#### Determination of Circulating Levels of Anti-SARS-CoV-2 Antibodies

Antibodies against the SARS-CoV-2 spike (S1) protein receptor-binding domain (S1-AB) encompassing all immunoglobulin classes (panIg) were determined using chemiluminescent immunoassay (ECLIA) (Elecsys Anti-SARS-CoV-2 S, Roche Diagnostics GmbH, Mannheim, Germany) on a COBAS 8000 analyzer (Roche, Basel, Switzerland) as prescribed by the manufacturer. Samples were measured at 10-fold dilution (Roche Cobas Universal Diluent) and re-measured at 400-fold dilution when exceeding the upper detection limit (250 U/mL). Results ≥0.80 U/mL are considered positive. PanIg antibodies against the SARS-CoV-2 nucleocapsid antigen (N-AB) were similarly determined with ECLIA (Elecsys Anti-SARS-CoV-2 assay, Roche, Basel, Switzerland) using a cut-off index based on positive and negative calibrators normalized to WHO standards [10]. Results presenting a ratio of signal/cut-off ≥1.0 are considered positive. 

### 2.3. Surrogate Assays for SARS-CoV-2-Neutralizing Activity

Virus neutralization activity of SARS-CoV-2 antibodies was measured with NeutraLISA (EUROIMMUN Medizinische Labodiagnostika AG, Lübeck, Germany) and cPass (GenScript Biotech, Piscataway, NJ, USA), which both measure binding of recombinant, biotin-labelled ACE2 receptor to recombinant SARS-CoV-2-S1/-receptor-binding domain immobilized on microtiter plates. Signals of ACE2 receptor bound in the presence of serum are inversely proportional to the neutralizing potency thereof. Following the manufacturers’ instructions, duplicate samples were processed on a semiautomatic ELISA processor by EUROIMMUN or using ELISA washer and reader from Tecan (Männedorf, Switzerland). Inhibition values (%) were derived from raw luminescence at 450 nm referenced to 620–650 nm and normalized to background without antibody. Cut-off values as provided by the manufacturers were ≥35% (positive) and <20% (negative) for the NeutraLISA, and ≥30% (positive) for the cPass.

### 2.4. Full Virus Endpoint Dilution Neutralization Test

Neutralization of entire SARS-CoV-2 virus B.1 isolate (Wuhan Hu-1 wildtype, GISAID accession number EPI_ISL_425126) served as reference assay for the neutralization capacity of sera. Two-fold serial dilutions (1:10 to 1:5120) of heat-inactivated sera (56 °C, 30 min) were prepared with maintenance medium (Dulbecco’s Modified Eagle Medium, Gibco (Waltham, MA, USA), Ref 11995-065, 100 U/mL Penicillin and 100 μg/mL Streptomycin, Gibco, Ref 11995-065, 2% Fetal Calf Serum, Pan Biotech, (Aidenbach, Germany) Cat P303031). A total of 50 µL of diluted serum samples was incubated (37 °C, 1 h) in 96-well cell culture TC plates (Sarstedt AG & Co. KG, Nümbrecht, Germany) with virus solution at an absolute TCID50 of 100. Subsequently, 100 µL of cell suspension containing 7 × 104 VERO cells/mL (ATCC-CCL-81, obtained from LGC Standards) was added to each sample, and incubation continued (37 °C, 5% CO_2_, 96 h). Subsequently, cytopathic effects (CPEs) were determined by microscopic inspection. The effective neutralization titre was defined as the highest CPE-negative sample dilution. Titres of ≥ 1:10 were considered positive. Controls included in each test series encompassed neutralization-negative and -positive serum samples (previously determined and stored at −20 °C), the effect of virus in the absence of serum, and growth controls of cells exposed neither to virus nor to serum.

### 2.5. Statistical Methods

IBM SPSS Statistics 28 software (IBM Corp. released in 2021. IBM SPSS Statistics for Windows, Version 28.0. Armonk, NY, USA: IBM Corp.) and Graph Pad Prism 9 (Graph Pad Software, Inc., San Diego, CA, USA: released in 2020. Graph Pad Prism 9 for Windows, San Diego, CA, USA: Graph Pad Inc.) were used for analysis. Normal distribution was tested according to Shapiro–Wilk and Q-Q graphs. Non-normally distributed data were descriptively analyzed by mean/median values, interquartile range and boxplots. Correlations were analyzed by Spearman correlation. Friedman’s test was used to detect differences between paired samples within a given dataset over time. Differences between infected and non-infected participants were analyzed by the Mann–Whitney-U test. Linear regression analysis was used to analyze relationships between vaccination responses at various timepoints. Correlation and effect size was assumed to be good at r ≥ 0.5 and moderate at r ≥ 0.3. For all tests, statistical significance was assumed at *p* < 0.05. Missing data (about 12%) were handled by listwise deletion.

## 3. Results

Samples obtained at each timepoint were tested for serum levels of antibodies against SARS-CoV-2 spike (S1) protein receptor-binding domain (RBD) (S1-AB) and antibodies against the SARS-CoV-2 nucleocapsid antigen (N) (N-AB). S1-AB served as a marker of infection as well as vaccination, whereas N-ABs served as a marker of infection only [11]. 

N-ABs are not expected to increase upon mRNA vaccination against spike (S1) protein receptor-binding domain (RBD) in the absence of infection. Consequently, N-AB-positive participants were assumed to have undergone asymptomatic infection with SARS-CoV-2 virus before vaccination or during the post-vaccination period monitored in the study. A total of 1.8% (n = 2) of the participants (n = 113, details: Appendix A) were positive for N-ABs and S1-AB in the sample obtained before the first vaccination, indicating that they had already undergone inapparent or unregistered infection(s) with the SARS-CoV-2 virus. Another participant tested positive for a SARS-CoV-2 infection according to a PCR test. According to the vaccination regimen of the institution, all participants were vaccinated irrespective of their serological state. However, in our study, N-AB- and PCR-positive participants were kept separate in statistical analyses. There were no N-AB-positive samples obtained six months after vaccinations, ruling out intercurrent inapparent SARS-CoV-2 infections.

S1-AB levels before and at various timepoints after vaccination were as follows: 99.1% (n = 109) of the N-negative participants (n = 110) and 100% (n = 116, details: Appendix A) tested positive after first and second vaccination, respectively. All N-negative participants monitored six months after the second vaccination (n = 95, details: Appendix A) were still S1-AB-positive. These values are in good agreement with vaccination responses observed elsewhere [12]. Mean values of S1-AB were 169 U/mL (0.4–1.004 U/mL) after the first vaccination and increased to 5704 U/mL (213–17.764 U/mL) after the second vaccination and dropped again to 1.019 U/mL (69–5.220 U/mL) six months after the second vaccination. A synoptic representation of S1-AB values obtained at the various timepoints of observation is given in Figure 1A.

Alterations in antibody levels over time were highly significant (*p* < 0.001), giving rise to highly inhomogeneous time courses of sero-responses (Figure 1B). Interindividual divergence started with immediate vaccination responses: certain participants showed a huge increase in antibody levels from an above-median level after the first vaccination to an even higher level above the median after the second vaccination (Index Pat. A), whereas other participants responded with sub-median rises in S1-AB to the first vaccination and exhibited no significant further increase following the second vaccination (Index Pat. B); see Figure 1B. Time courses of S1-AB levels during the six months after the second vaccination were even more heterogeneous, encompassing a drop to as low as 2.6% (250 U/mL of 9.724 U/mL, Index Pat. C) as well as maintenance of as much as 67.5% (5.220 of 7.725 U/mL, Index Pat. D) of the initial S1-AB level reached after the second vaccination. Differences in immediate and long-term humoral vaccination responses exhibited no significant correlation with age, gender, or any known co-pathologies of the study participants. Most notably, S1-AB levels immediately after the second vaccination exhibited only a very moderate correlation (r = 0.54, *p* < 0.001) with corresponding residual S1-AB levels observed six months later. The rather poor linear regression of those data (r^2^ = 0.16, *p* < 0.001) suggests that immediate humoral vaccination response and long-term maintenance of humoral immunity are not stringently linked in quantitative terms. (Figure 1C).

Samples of participants having undergone SARS-CoV-2 infection before vaccination (n = 3, details: Appendix A) were identified by increased serum levels of N-ABs and/or a positive PCR result. These samples exhibited many-fold higher levels of S1-AB. After the first vaccination, the mean value of S1-AB in post-infection samples was 47,738 ± 3.002 U/mL as opposed to 169 ± 16.6 U/mL in non-infected samples. After the second vaccination, the mean value of S1-AB in post-infection samples was 43,001 ± 1.532 U/mL as opposed to 5.704 ± 322.9 U/mL in non-infected samples. These differences were highly significant (*p* < 0.001). In the long run, the augmenting effect of SARS-CoV-2 infection on vaccination response started to diminish. At six months after the second vaccination, the mean value of S1-AB in post-infection samples was 3.070 ± 417 U/mL as opposed to 1.019 ± 88.5 U/mL in non-infected samples. This difference was still significant (*p* = 0.001) but quantitatively less pronounced than at the timepoints directly after vaccination (see Figure 1A and Appendix A).

In the next step, we compared S1-AB serum levels with corresponding virus-neutralizing activity of the sera. For that purpose, all samples were probed for their potency to inhibit the binding of biotin-labelled ACE2 receptor to immobilize recombinant SARS-CoV-2-S1/-RBD (NeutraLISA, EUROIMMUN, Lübeck, Germany), which is considered a practical diagnostic surrogate for the neutralization of cytopathic effects of the full viable virus as determined in cell culture. After the first vaccination and six months after the second vaccination (see Appendix A), levels of S1-AB correlated strongly with the corresponding virus neutralization capacity of the sera (r2 = 0.774 to 0.845). Immediately after the second vaccination, a similar analysis was determined not to be meaningful since the upper measuring limit of the NeutraLISA at 100% was already attained by sub-median levels of S1-AB. Thus, the limited dynamic range rendered the NeutraLISA uninformative in the situation of recent re-immunization. Similar results were obtained by cPass (not shown). The two surrogate assays exhibited excellent linear correlations across all timepoints (r^2^ = 0.774 to 0.932, *p* < 0.001) (Appendix A). It should be noted that the cPass assay appeared slightly more sensitive in the low range (after the first vaccination) but yielded similar values (around 98%) after the second vaccination. 

In summary, the two surrogate assays for virus neutralization capacity failed to provide meaningful additional information regarding immediate vaccination responses. However, they may be useful in long-term monitoring of humoral vaccination responses. To follow up on the latter notion, NeutraLISA data obtained at six months after the second vaccination was scrutinized for relevance. Based on comparisons with WHO standards and a full virus endpoint dilution neutralization test (full virus NT), inhibition values of ≥35% obtained by the NeutraLISA in post-infection sera are proposed to indicate effective virus neutralization potency [13]. However, according to our own unpublished observations, the neutralization potency of antibodies induced by S1-spike protein-directed vaccination may be overestimated by these surrogate assays as compared to the results obtained with the full virus NT, which is currently considered the reference assay. To follow up on this notion, samples collected six months after the second vaccination were re-tested with a full virus NT. For 95 samples, interpretable results were obtained. Within these samples, the surrogate assays showed strong correlations with the full virus NT (r^2^ = 0.79, *p* < 0.001 for NeutraLISA, r^2^ = 0.77, *p* < 0.001 for cPass) (Figure 2A,B), which confirms the results of previous studies [13]. However, in the low range, positive–negative discrimination by the surrogate assays did not sufficiently match the results of the full virus NT. Most notably, the surrogate assays yielded a significant number of false-positive results (5/89 in both tests) (Figure 2A,B, inserts), suggesting that they may not be a safe companion diagnostic for long-term monitoring of vaccination with mRNA-based vaccines such as Spikevax (Moderna, Cambridge, MA, USA). 

Consequently, we addressed the question of which other diagnostic tools or staged strategies could possibly improve the safety of serologic monitoring of long-term vaccination responses. First, we investigated whether a full virus NT titre ≥10 at six months after vaccination could possibly be predicted from the quantitative levels of S1-AB measured either directly (see Appendix A) or six months after second vaccination. S1-AB levels measured directly after the second vaccination were poorly correlated with the full virus NT obtained six months later (r^2^ = 0.54, *p* < 0.001), which was expected given the equally poor correlation with quantitative S1-AB determined six months later (Figure 1C). However, S1-AB levels measured six months after the second vaccination exhibited a reasonably strong correlation with neutralizing capacity as determined by the full virus NT at the same time (r^2^ = 0.79, *p* < 0.001) (Figure 3A), allowing for determining a cut-off at 1000 U/mL to discriminate a major portion (35/89 of the full virus NT-positive samples from all full virus NT-negative samples (Figure 3A, insert). Incidentally, the fraction above that cut-off encompassed all samples having undergone infection in addition to double vaccination (Figure 3B, black circles). 

The remaining 63/89 samples below the cut-off (i.e., exhibiting S1-AB levels <1000 U/mL six months after second vaccination) (Figure 3B, symbols below dashed line) encompassed all 6 full virus NT-negative samples (Figure 3B, closed red circles) but also 57 full virus NT-positive samples (Figure 3B, open circles below dashed line). In search of a practical diagnostic tool allowing us to discriminate within this group between NT-negative and -positive samples, we reassessed the corresponding results of the surrogate assays for virus neutralization. Upon re-adjusting the cut-off level of NeutraLISA and cPass to 64 and 72%, respectively, it was thereby possible to discriminate between 5/6 true full virus NT-negative samples within the samples having S1-AB levels <1000 U/mL (Figure 4). 

In summary, the staged diagnostic strategy applied six months after the second vaccination detected five out of six full virus NT-negative samples, i.e., it had a sensitivity for a presumably insufficient virus neutralization capacity of 83.3%. As few as 14/89 (using NeutraLISA) or 6/89 (using cPass) were thereby falsely classified as virus NT-negative, i.e., corresponding specificity values were 84.2 and 96.2% for NeutraLISA and cPass, respectively.

## 4. Discussion

### 4.1. Rationale and Aim

Currently, SARS-CoV-2 vaccinations follow fixed temporal schedules prescribed by the manufacturers of the vaccines that are corroborated by guidelines and recommendations of national and international health agencies [14,15,16,17]. The ongoing appearance of new virus mutants and the increasing incidence of SARS-CoV-2-infection and COVID-19 disease in vaccinated people [18,19,20] demonstrate that vaccinations often fail to convey permanent immunity, and regular re-vaccinations are to remain a necessity in routine health care [21]. 

The rigidly scheduled regimen of re-vaccination currently employed to break the pandemic waves appears to be safe in terms of undesired side effects [22]. However, a more flexible vaccination strategy may have to be adopted eventually for the following reasons: (i) the duration of protection conveyed by current mRNA- and vector-based vaccines differs considerably [23]; (ii) the heterogeneity of duration of vaccination protection will further increase as protein- and whole-virus-based types of vaccines (such as Nuvaxovid (Novavax, Gaithersburg, MD, USA)) are introduced [24]; (iii) individual SARS-CoV-2 immunity and COVID-19 protection of vaccinated people is bound to vary even more due to unknown re-immunization by asymptomatic SARS-CoV-2-infections [25,26]; and (iv) ultimately, synchronous pandemic infection waves will lead to continuous asynchronous endemic re-infection, making it even more difficult to select optimal timepoints for re-vaccination [27,28]. 

In summary, the above arguments suggest that a rigidly scheduled regimen for SARS-CoV-2 vaccination may soon become obsolete. Instead, it may become necessary to adapt re-vaccination to the individual immune status. This expectation implies the need to gauge the individual state of SARS-CoV-2 immunity using a diagnostic test [29]. Currently, there is no analytical correlate of protection against SARS-Cov-2 infection or against COVID-19 disease. Nevertheless, the question of which companion diagnostics may possibly be suitable to support an individualized vaccination strategy is raised. 

To address this question, here we have here evaluated several tests for humoral SARS-CoV-2 immune responses, which are currently commercially available and practical in the setting of routine health care diagnostics [30,31,32]. We have investigated which of these tests could be used to monitor the waning of vaccination. In addition, we have compared our results to the full virus NT, which is not practical in routine health care but considered the serological test most closely reflecting humoral immunity [33,34]. 

### 4.2. Salient Findings

Humoral vaccination responses exhibited a huge interindividual heterogeneity in the study collective in terms of (i) maximal serum levels of S1-AB induced by vaccination, (ii) time courses of S1-AB levels over six months and (iii) residual S1-AB levels after six months. These observations are in line with other studies [35].Immediate response and long-term maintenance of vaccination-induced antibodies were not stringently linked in quantitative terms, precluding judgement of durability of vaccination response from antibody levels measured shortly thereafter.Four types of time courses of vaccination response could be identified: (i) high initial response followed by rapid decline, (ii) middling initial response followed by slow decline, (iii) middling initial response followed by fast decline and (iv) low initial and overall response. Types (iii) and (iv) tended to result in sub-average S1-AB levels after six months and were found in about half of the participants.Surrogate assays for gauging the vaccination-induced serological potential of virus neutralization failed to provide meaningful information shortly after vaccinations due to limitations of measuring range and upper measuring limits.At six months after vaccination, the serological potential of virus neutralization tended to be overestimated by surrogate assays as compared to the full virus NT, supporting previous notions that indiscriminate use of these assays would not provide adequate warning of crucial waning of immunity [36].Lack of functional virus protection (as defined by full virus NT negativity) can possibly be detected during prolonged waning periods by a staged strategy employing surrogate assays to detect S1-AB levels below a cut-off of <1000 U/mL, which were judged by an elevated cut-off of around 70%. At six months after the second vaccination, virus NT-negative samples could thus be detected with a sensitivity of >80% and a specificity of between 80 and 90%, depending on which surrogate assay was used.

### 4.3. Limitations

All participants were vaccinated with Spikevax (Moderna Biotech), which rendered the study collective homogenous and produced significant results for this specific kind of vaccine. Our findings cannot be readily applied to other vaccines, especially protein- or vector-based ones.Our serological tests were not adapted to virus mutants. Thus, our results apply only to the initial SARS-CoV-2 Wuhan virus isolate. Existing SARS-CoV-2 vaccines have been found to be less efficient against Delta, Omicron and other SARS-CoV-2 variants of concern (VOCs) [18]. Consequently, application of the proposed strategy to assess vaccinations against such VOCs will require adaption of the immunogenic assay target and re-evaluation of sensitivities, specificities and cut-offs.Having included only three infected people in our study, we cannot add significantly to previous studies comparing antibody levels in persons with or without SARS-CoV-2 infection before and during vaccination on a larger scale [31,37]. However, we clearly show that the proposed diagnostic strategy worked similarly for vaccinated persons with and without infection.The full virus NT is currently considered as the ex vivo test that most closely reflects functional humoral immune response [34]. However, it is not a direct measure of immunity itself, and it remains unknown how these data are related to real-life immunity. The same limitation applies to our data, which have been calibrated to the full virus NT. Yet, it is not fully understood which serological or cellular parameter is a direct correlate to sufficient immune protection. Therefore, measuring neutralizing antibodies can indicate the necessity for re-vaccination but cannot be understood as a strict recommendation.

### 4.4. Concluding Remarks

Current epidemiologic studies predict that regular re-vaccination against the SARS-CoV-2 virus will become a future necessity in routine health care [21], which poses a number of challenges: even a single type of COVID-19 vaccine exhibits considerable interindividual variability regarding levels and persistence of humoral immune responses thereby induced (shown here and in previous studies [38,39]). Variance of immune responses further increases when several types of vaccines are in play [23]. Moreover, intercurrent (possibly asymptomatic) infections have to be taken into account, since undesired vaccination effects tend to be more severe following recent infection [40]. In conclusion, it is to be expected that re-vaccination will soon have to be tailored to individual immune status. Serological surveillance of individual SARS-CoV-2 immunity will become even more important when synchronous pandemic infection waves lead to continuous asynchronous endemic re-infection. The diagnostic strategy proposed here may be useful in facing the above challenge, as it provides a reliable way of gauging levels and functionality of circulating SARS-CoV-2 antibodies, which so far requires ex vivo tests of virus neutralization in cell culture impracticable in routine health care [41]. Strictly speaking, the data presented here are already superseded, because they are only valid for the determination of antibody activity against the original Wuhan type of SARS-CoV-2. However, following re-calibration of the immunological tests and their cut-offs, the procedure can readily be applied to any virus mutant dominating the COVID-19 endemic in the future.

## Figures and Tables

**Figure 1 vaccines-10-01044-f001:**
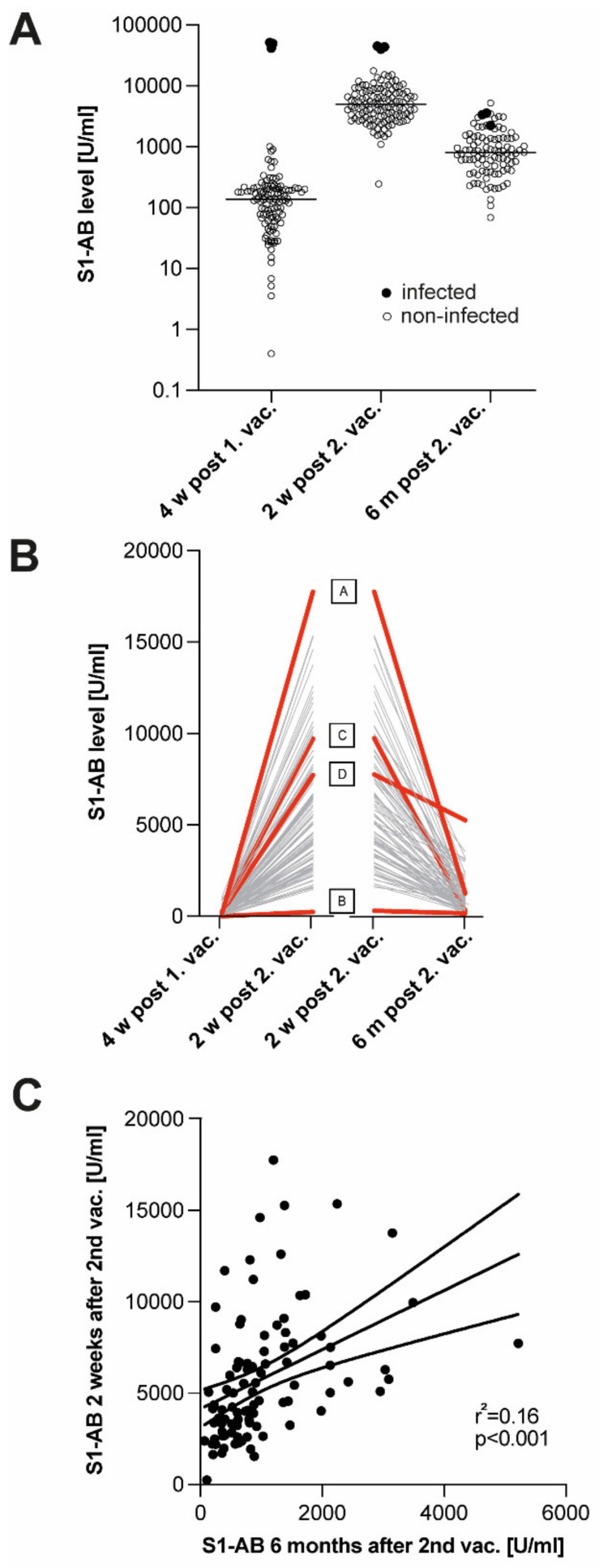
Serum levels of S1-AB after vaccination with COVID-19 vaccine Spikevax. (**A**) S1-AB levels measured four weeks after first (**left**), two weeks after second (**middle**) and six months after second vaccination (**right**). Median values indicated by horizontal bars. Brackets indicate significant (*p* < 0.001) differences of the means. Closed symbols: SARS-CoV-2 infection prior to vaccination. (**B**) Developments of S1-AB serum levels between first and second vaccination (**left**) and 6 months after second vaccination (**right**). Exemplary time courses are highlighted (red) and indexed (capitals). (**C**) Correlation of S1-AB serum levels at two weeks and six months after second vaccination. Linear regression of the data and 95% confidence interval indicated by solid and dashed lines, respectively (r^2^ = 0.16, *p* < 0.001). For numbers of included participants, see Appendix A.

**Figure 2 vaccines-10-01044-f002:**
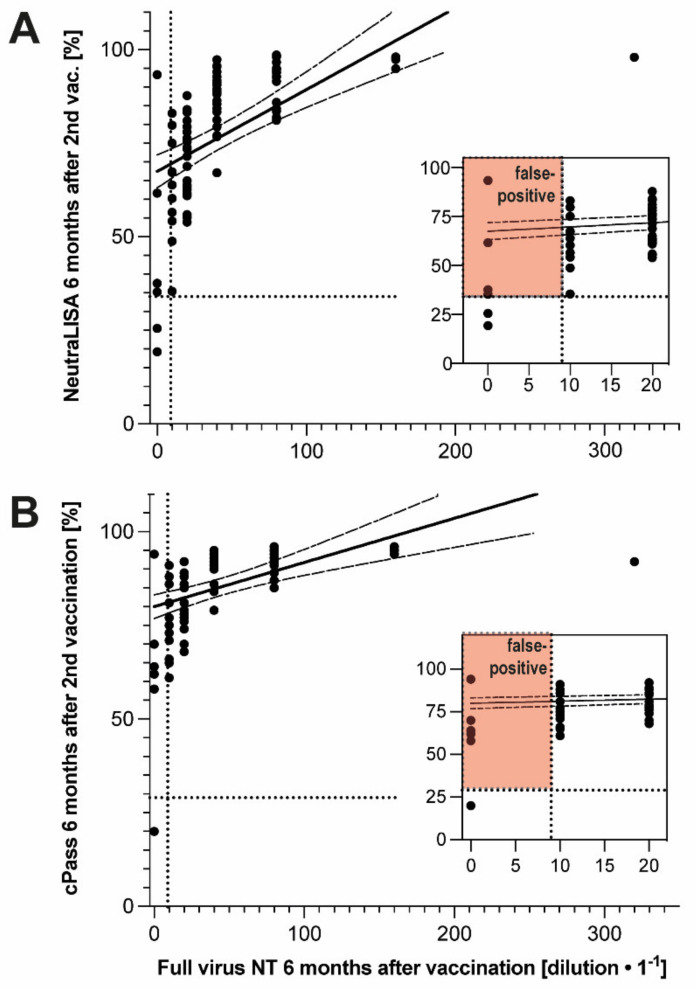
Virus neutralization capacity six months after second vaccination, as determined by surrogate assays and full virus NT. (**A**) Correlation of NeutraLISA (Lübeck, Germany) with full virus NT six months after second vaccination; linear regression of the data and 95% confidence interval indicated by solid and dashed lines, respectively (r^2^ = 0.79, *p* < 0.001); insert: blow ups of low-level range. (**B**) Correlation of cPass with full virus NT six months after second vaccination; linear regression of the data and 95% confidence interval indicated by solid and dashed lines, respectively (r^2^ = 0.77, *p* < 0.001); insert: blow up of low-level range. For numbers of included participants, see Appendix A.

**Figure 3 vaccines-10-01044-f003:**
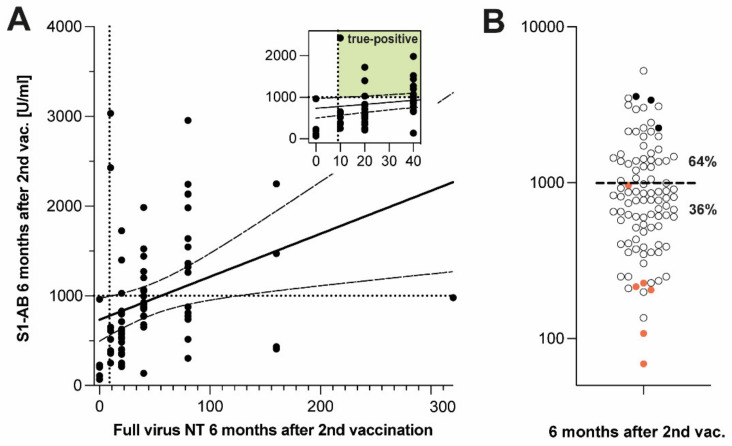
Levels of S1-AB and titres of full virus NT six months after second vaccination. (**A**) Comparison of S1-AB and full virus NT six months after second vaccination; linear regression of the data and 95% confidence interval indicated by solid and dashed lines, respectively (r^2^ = 0.79, *p* < 0.001); insert: blow up of low-level range. (**B**) S1-AB levels six months after second vaccination; horizontal dashed bar: cut-off for full virus NT negatives. Percentages: fractions of samples located above and below cut-off. Closed symbols (black): SARS-CoV-2 infection prior to vaccination, closed symbols (red): full virus NT negatives. For numbers of included participants, see Appendix A.

**Figure 4 vaccines-10-01044-f004:**
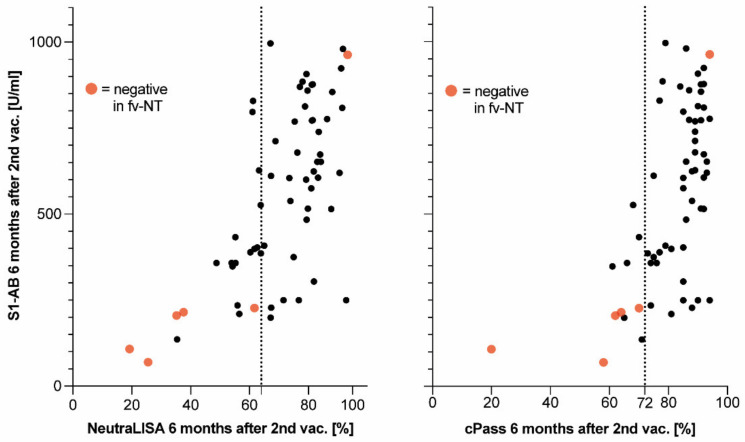
Adjustment of surrogate assays’ cut-offs for the discrimination of full virus NT-negative samples with S1-AB levels <1000 U/mL at six months after second vaccination compared with corresponding values of NeutraLISA (**left**) and cPass (**right**). Dashed lines: optimized cut-offs of surrogate assays for discrimination of full virus NT-positive from full virus NT-negative samples. For numbers of included participants, see Appendix A.

## Data Availability

Not applicable.

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
