# Peer review of "A Diagnostic Strategy for Gauging Individual Humoral Ex Vivo Immune Responsiveness Following COVID-19 Vaccination"

_vaccines, 2022, doi:10.3390/vaccines10071044_

Round 1

Reviewer 1 Report

The manuscript by Kuechler and colleagues describes a study that aimed to investigate the antibody responses to SARS-CoV-2 infection following vaccination with two doses of the Spekevax vaccine (Moderna) in hospital employees. The authors propose a strategy to use ELISA-based analyses as a surrogate to laborious standard neutralization assays in order to measure neutralizing antibodies (NT-abs) against SARS-CoV-2.

It is well established that antibody levels including those antibodies that neutralize the virus vary considerably following SARS-CoV-2 vaccination. Both assays to measure total antibody responses as well as surrogate assays for NT-abs are available, still there is a level of uncertainty as to whether these assays reflect true NT-abs activity in the sera of either vaccinees or infected individuals. Studies to investigate the performance of assays for routine diagnostics to measure NT-abs responses are welcome for the monitoring of both vaccine responses as well as of antibody kinetics following infection.

The manuscript provides some interesting data with respect to the different kinetics of NT-abs responses following vaccination against SARS-CoV-2. However, there are some issues that are critical to the manuscript.

1.            At the present time there is no correlate of protection, neither against infection nor against COVID-19. Thus the title is misleading, as the NT-abs levels that are measured in this manuscript are not linked (yet) to the “immune state”.

2.            Lines 64 following. Considering the infection rates in vaccinated individuals, it is highly questionable if there is significant immune protection against SARS-CoV-2 infection. There is clear protection against disease. This should have been clearly differentiated throughout the manuscript.

3.            A general criticism relates to the statement in the manuscript that NT-abs titers may be used to monitor vaccine effectiveness. Unfortunately, we do not know yet which, if any, parameter may provide a measure of protection against COVID-19 (as mentioned there may be no such immunity against mere SARS-CoV-2 infection). The establishment and validation of routine methodologies to measure NT-abs are certainly helpful. But as long as there is no correlate of protection established, levels of NT-abs cannot be used to guide vaccination. The manuscript would have to be revised to clearly emphasize this point.

Minor.

-          Line 68. The term “functional humoral immunity” is not well defined, as there may be other antibodies without neutralizing activity which may display antiviral function. Thus if that term is used it should be clearly defined as to what is meant at the first mentioning.

Reviewer 2 Report

The paper highlights an important aspect of vaccination strategy, as mentioned by the Authors the data here presented have been superseded by new variants, nevertheless the data show has immune system of healthy subjects reacts to Covid vaccination

The Authors should clarify if there is any correlation between vaccine response with age, gender or co-pathologies of the patients (i.e. diabetes, hypertensoin, etc..). Moreover as the patients were tested six month apart  the second vaccination, in a pandemic period, it should be clarified if a intercurrent SarsCov2 infection has been ruled out.

Moreover it could be interesting correlate the SarsCov2 disease severity with immune response if someone of the patients acquired the infection in the follow up.
